# Characteristics of Koji Using Liquid Starter for Soy Sauce Production

Jonghoon Choi [1,†], Taeeun Kwon [2,†] , Yeongbin Park [1] and Augustine Yonghwi Kim [1,*]

[1] Department of Food Science and Biotechnology, Sejong University, 209 Neungdong-ro, Gwangjin-gu, Seoul 05006, Republic of Korea; jh1056@hanmail.net (J.C.)
[2] Carbohydrate Bioproduct Research Center, Sejong University, 209 Neungdong-ro, Gwangjin-gu, Seoul 05006, Republic of Korea; kwonsh80@sejong.ac.kr
[*] Correspondence: kimyh@sejong.ac.kr; Tel.: +82-2-3408-3228; Fax: +82-2-3408-4319
[†] These authors contributed equally to this work.

**Abstract:** Soy sauce is a widely consumed seasoning derived from soybeans and wheat. This study explored the application of innovative techniques to enhance the traditional soy sauce preparation process. Fungi were isolated from a commercial koji starter, and the *Aspergillus oryzae* strain BJ-1 was identified. Additionally, an examination of the methods to optimize the medium composition for liquid starters revealed the impact of varying the medium composition on mycelial growth and enzyme activity. Specifically, compositions containing >10% defatted soybean meal and wheat in a 55:45 ratio resulted in elevated mycelial growth and enzymatic activity, making them promising candidates for koji production. The effect of different inoculation ratios of liquid starter on the characteristics of koji was also investigated, and a 10% inoculum was found to be preferable because of its advantageous characteristics of enzyme activities and pH for soy sauce production. This study contributes to the enhancement of the efficiency and safety of soy sauce production through innovative liquid culture techniques.

**Keywords:** soy sauce; koji production; liquid culture; *Aspergillus oryzae*; enzymatic activity; fermentation





## 1. Introduction

Soy sauce is a widely consumed seasoning prepared from soybeans, wheat, and saltwater, and it can be categorized into two types based on its manufacturing process: biological and chemically hydrolyzed soy sauce [1,2]. Biological hydrolysis involves three steps: the pretreatment of raw materials, koji preparation, and the Moromi process [3]. During pretreatment, the raw materials are steamed or roasted [4]. In the koji preparation step, a mixture of pretreated raw materials is inoculated with spores of a koji-making fungus, such as *Aspergillus oryzae*, and incubated for three days under controlled temperature and humidity conditions [5]. In the Moromi process, koji is mixed with saline and aged for six months [6]. The combination of a semi-anaerobic environment created underwater and high salt content reduces fungal growth but facilitates the continuous hydrolysis of raw materials, thereby increasing the content of soluble materials [7–9]. The enzymes produced from koji breakdown proteins, carbohydrates, and lipids, and the substrates degraded by fungal enzymes during koji preparation, are eventually used for the Moromi maturation stage by other microorganisms (lactic acid bacteria, yeast, etc.). Koji processing substantially influences soy sauce flavor, making the koji preparation process critical to the quality of the final product [10,11].

*A. oryzae* is the most commonly used inoculum for koji production, typically at approximately $2 \times 10^8$ colony-forming units (CFUs)/g spore, for inoculation into mixtures of soybeans or defatted soybean meal and wheat under conditions of controlled temperature (28–36 °C) and high relative humidity (>90%) [12]. *A. oryzae* produces major enzymes such as proteases and amylases, which digest proteins and carbohydrates in koji [13]. In addition,

various other enzymes, including pectinases, cellulases, and hemicellulases, digest the cell wall structure of soybeans and/or wheat, thereby limiting culture preparation. The fungal enzymes greatly increase koji's characteristics by hydrolyzing proteins, carbohydrates, and lipids into amino acids, monosaccharides, and fatty acids, respectively [14]. These koji products and their relative contents usually affect the quality of soy sauce [15]. Solid fermentation has traditionally been used for koji production [16]. For industrial soy sauce production, powdered spores of *A. oryzae* mixed with pretreated low-moisture raw materials are used as an inoculum for koji production via solid fermentation. However, traditional solid fermentation using spores has some disadvantages, including susceptibility to bacterial contamination, leading to mycotoxin production or spoilage [17]. Additionally, koji production rooms require careful maintenance of their temperature, humidity, and oxygen, making environmental control difficult [18].

Fungal liquid culture offers the advantages of the fast and cost-effective mass production of mycelia and specific useful components [19]. This process is relatively safe from bacterial contamination during liquid culture inoculation and can be easily controlled to facilitate the recovery of mycelia and extracellular enzymes [20]. However, liquid culture has not yet been used for industrial koji production because of its weaker enzymatic activity than that of solid-state (surface) cultures [21]. The predominantly formed proteases are sporogenous, and spores are not formed in liquid culture, which affects enzymatic activity [22]; therefore, liquid inocula have not been used in industrial koji production. Research on the production of flavor components during Moromi fermentation and maturation using liquid koji remains insufficient [23].

In this study, we investigated the feasibility of using liquid culture inocula for koji production, a critical process in soy sauce production. We conducted an experiment to compare the use of a liquid culture starter with a traditional spore-based control. The use of a liquid starter enhances the efficiency and manageability of this process, particularly in commercial soy sauce production. Moreover, maintaining a pure culture within a closed environment can substantially reduce the risk of bacterial contamination and the presence of microorganisms capable of producing harmful substances such as mycotoxins, thus ensuring a cleaner and safer soy sauce production process. Furthermore, workers exposed to relatively high health risks owing to handling spores [24] can be protected from lung diseases. In the present study, the optimal conditions for producing a liquid starter suitable for koji production were established. The possibility of koji production by *Aspergillus* spp. using liquid culture in a closed system holds important potential benefits, as the fermentation and ripening of soy sauce in a closed system eliminates the need for substantial quantities of salt traditionally used in the Moromi production stage and eventually allows for low-salt soy sauce production. This not only shortens the ripening period by promoting microbial growth, but also results in the production of salt-free fermented soy sauce, which can contribute significantly to advances in the food industry.

## 2. Materials and Methods

### 2.1. Isolation and Identification of Fungi from Koji Starter

The koji starter 'MC-01' (Nihon Jyozo Kogyo Ltd., Tokyo, Japan) was used in this study. A total of 1 g of koji starter 'MC-01' was homogenized in 9 mL of 0.1% (*w/v*) Tween 80 (Sigma-Aldrich, St. Louis, MO, USA) solution and serially diluted up to $10^{-8}$. One milliliter of each dilution was spread on Potato Dextrose Agar (PDA; Difco™, Becton, Franklin Lakes, NJ, USA) and incubated at 28 °C for three days. A single colony was selected and sub-cultured in PDA using a loop (1 μL; SPL Life Science, Pocheon, Republic of Korea) and cultured in 100 mL of Potato Dextrose Broth (PDB; Difco™, Becton, Franklin Lakes, NJ, USA). After culturing at 28 °C for three days, the culture was ground twice for 30 s using a sterile grinder (Waring International, New Hartford, CT, USA) to prevent pellet growth. The homogenized culture was mixed with 25% (*v/v*) glycerol (Daejung Chem., Siheung, Republic of Korea) to prepare the stock culture, which was stored at −80 °C and used to

prepare liquid starters. Strains were identified via 18S rRNA sequencing (Macrogen, Seoul, Republic of Korea).

## 2.2. Enzyme Activities and Growth Patterns According to the Medium Compositions in Liquid Starters

Defatted soybean meal and wheat, imported from the United States, were obtained from Maeil Foods Co., Ltd. (Suncheon, Republic of Korea). The proximate compositions of defatted soybean meal and wheat were determined using the AOAC 21st edition [25], and both were pretreated before use. Defatted soybean meal was soaked to 1.3 times its weight in water for 15 min and steamed at 121 °C for 50 min, and 250 g of wheat was roasted at 200 °C for 17 min. The pretreated defatted soybean meal and wheat were finely ground using an SMX-C2000WK blender (Shinil Electric Mixer, Seoul, Republic of Korea) and passed through a 30-mesh sieve before being added to the medium. A 100 mL medium was prepared with concentrations of defatted soybean meal and wheat (mixed at a ratio of 55:45) at 5%, 10%, and 15% in deionized water. A stock culture of fungi isolated from the koji starter was pre-cultured in PDB at 28 °C for 48 h. The culture was then inoculated in 10% medium. The liquid starters were incubated at 28 °C and 150 rpm for 96 h, and samples were collected at 0, 24, 48, 72, and 96 h to determine the optimal medium conditions for the liquid starter. Ten grams of each sample was washed twice with distilled water and filtered through a 70-mesh sieve. The unfiltered solids were dried at 105 °C for 24 h, and the cell mass was analyzed by measuring the dry weight. Samples were centrifuged at $8000 \times g$ and 4 °C for 20 min (Himac CR21G, Hitachi Koki Co., Ltd., Tokyo, Japan). The supernatants were filtered using a 0.45 μm syringe filter. The glucose content, pH, and acidity were measured and used as crude enzyme solutions for enzyme assays. All experiments were repeated at least thrice. Glucose content was measured using a YSI 2700 Select Biochemistry Analyzer (YSI Inc., Yellow Springs, OH, USA). pH was measured using a digital pH/ion meter (DP 215M, DMS, Gimhae, Republic of Korea), and the acidity was assessed by quantifying the lactic acid content. To measure the acidity, 1 g of the sample was diluted to 250 mL with distilled water and filtered through Whatman No. 2 Filter Paper. The filtrate (10 mL) was titrated to pH 8.3 with 0.1 N sodium hydroxide (Samchun Chemical Co., Ltd., Seoul, Republic of Korea). The acidity was calculated as follows:

$$\text{Acidity (\%)} = V \times F \times A \times D \times 1/S \times 100$$

V = Titration volume of 0.1 N sodium hydroxide (mL).
F = Factor of 0.1 N sodium hydroxide.
A = Amount of organic acid equivalent to 1 mL of 0.1 N sodium hydroxide (lactic acid: 0.0090).
D = Dilution factor.
$S$ = Amount of sample (g).

## 2.3. Characterization of Koji by Assessing Inoculation Rate of Liquid Starter

Figure 1 shows a flowchart of the koji fermentation process using two different starter cultures. Defatted soybean meal and wheat were pretreated as described above and mixed in a 55:45 ratio. Koji was produced from these mixtures and inoculated with MC-01 spores and liquid starters derived from the koji starter MC-01. The spores were inoculated at $2 \times 10^8$ CFUs/g, and the liquid starters were inoculated at 1%, 5%, 10%, or 20% of the mixture weight. The inoculated mixture was transferred to a mesh tray covered with cloth and placed in an incubator at 28 °C and 99% relative humidity. The koji was hand-mixed for 18, 28, and 44 h. After mixing, each type of koji was collected at 0, 24, 48, and 72 h, and enzyme activities were assessed. One gram of each sample was diluted 10-fold with deionized water. Crude enzymes were extracted via sonication at 50 °C for 1 h, followed by centrifugation at $8000 \times g$ and 4 °C for 20 min. The supernatants were filtered using a 0.45 μm membrane syringe filter and used as crude enzyme solution in the enzyme assay. All experiments were repeated three times.

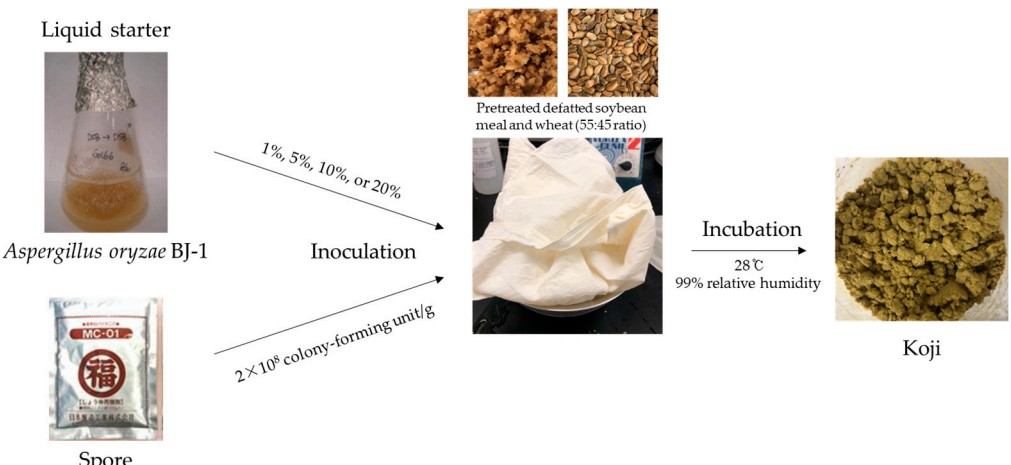

**Figure 1.** Flowchart of koji fermentation processes using two different starter cultures.

*2.4. Determination of Enzyme Activities*

2.4.1. α-Amylase Activity

α-amylase activity was determined based on the amount of glucose released during starch degradation. The substrate used was 1% (*w*/*v*) soluble starch (Loba Chemical Co., Ltd., Mumbai, India) solution in distilled water. For the assay, 1 mL of the sample was mixed with 1 mL of starch substrate and incubated at 40 °C for 20 min. The enzymatic reaction was terminated by adding 3 mL of 1.5 M trichloroacetic acid (Samchun Pure Chemical Co., Ltd., Seoul, Republic of Korea). The glucose content in the resulting mixture, which was filtered using a 0.45 μm membrane syringe filter, was measured using a YSI 2700 Select Biochemistry Analyzer. One unit of α-amylase activity was defined as the amount of enzyme capable of liberating 1 μmoL of glucose under these conditions.

2.4.2. Total Endo-Protease Activity

The taste of soy sauce arises from an increase in the total nitrogen concentration in the soluble fraction owing to the breakdown of soybean protein and the subsequent liberation of amino acids or peptides. Endopeptidases hydrolyze proteins and release peptides. A recent study reported that short peptides enhance soy sauce flavor [26]. To measure total endo-protease activity, azo-casein (S-AZCAS 12/07, Megazyme, Bray, Ireland) was used as a substrate, and a modification of the endo-protease azo-casein assay described in Megazyme was employed. The substrate solution was prepared by dissolving 2 g of azo-casein in 4 mL of ethyl alcohol (99.9%, ACS grade; J.T. Baker, Phillipsburg, NJ, USA) and adding 96 mL of 100 mM sodium phosphate buffer (pH 7.0) for 10 min. One milliliter each of the sample and substrate solution was reacted at 40 °C for 10 min, and 3 mL of 1.5 M trichloroacetic acid was added to stop the reaction. After 5 min of incubation at room temperature, the mixture was filtered through Whatman No. 1 Filter Paper. One milliliter of the filtrate was transferred to a cuvette, and the absorbance was measured at 440 nm using a UV-vis spectrophotometer (DU650; Beckman Coulter Inc., Brea, CA, USA) at the Biopolymer Research Center for Advanced Material (BRCAM). Protease activity was calculated as follows:

$$\text{Protease (mUnit/mL)} = 146 \times \text{Abs (440 nm)} - 4$$

2.4.3. Acidic Protease Activity

Protease activity was assayed using the method described by Uchida [27] with modifications. Acidic protease activity was evaluated at pH 4.0 by measuring the absorbance at 660 nm of the substance released from a casein solution that had reacted with Folin–Ciocalteu's reagent (Sigma-Aldrich Co., St. Louis, MO, USA). Subsequently, 0.1 mL of the crude enzyme was added to 5 mL of 0.6% casein solution in 50 mM phosphate

buffer (pH 5.0), followed by incubation at 30 °C for 10 min. The enzymatic reaction was terminated via the addition of 5.0 mL of a reagent consisting of 0.11 M trichloroacetic acid, 0.33 M acetic acid, and 0.22 M sodium acetate. Next, the mixture was filtered through Whatman No. 6 Filter Paper, and 2 mL of the filtrate was combined with 5 mL of 0.55 M sodium carbonate and 1.0 mL of 0.6 N Folin–Ciocalteu's reagent. This mixture was allowed to react at 30 °C for 30 min, and the absorbance was measured at 660 nm. One unit of enzyme activity was defined as the amount of enzyme that liberated 1 g of tyrosine per min under these conditions.

### 2.4.4. Neutral Protease Activity

Neutral protease activity was assayed using the same method used for acidic protease [27]. The substrate (0.6% casein) was diluted in 50 mM phosphate buffer (pH 7.0). One unit of enzyme activity was defined as the amount of enzyme that liberated 1 g of tyrosine per min under these conditions.

### 2.4.5. Alkaline Protease Activity

Alkaline protease activity was determined as previously described for acidic protease [27]. The substrate (0.6% casein) was diluted in 50 mM ammonia buffer (pH 8.0). One unit of enzyme activity was defined as the amount of enzyme that liberated 1 g of tyrosine per min under these conditions.

### 2.4.6. Glutaminase Activity

Glutaminase level was measured as the amount of ammonia released from L-glutamine [28]. Specifically, 0.5 mL of 0.04 M L-glutamine (Daejung Chemicals & Metals Co., Ltd., Siheung, Republic of Korea) was prepared in 0.1 M phosphate buffer (pH 7.0), and the sample (0.5 mL) was mixed and incubated at 37 °C for 30 min with agitation at 150 rpm in a shaking incubator. To stop the reaction, 1 mL of 1.5 M trichloroacetic acid was added, and 0.1 mL of this mixture was diluted with 3.7 mL of distilled water. Nessler's reagent (0.2 mL) was added to induce color development. The absorbance was measured at 450 nm, and a standard curve was constructed using ammonia sulfate (ACS reagent, Sigma-Aldrich). The glutaminase activity was calculated by substituting the absorbance value of the sample into the equation derived from the standard curve. One unit of glutaminase activity was defined as the amount of enzyme capable of liberating 1 μmoL of ammonia under these conditions.

### 2.5. Statistical Analysis

All experimental results were analyzed via analysis of variance using SPSS (ver. 24; IBM Inc., Armonk, NY, USA). When the main effect was statistically significant ($p < 0.05$), Duncan's multiple range test was applied.

## 3. Results and Discussion

### 3.1. Isolation and Identification of Fungi from Commercial Koji Starter

A single fungal colony was inoculated in PDA, and the microorganisms were incubated and observed. White colonies were formed during the early stages of culture, which changed from cloudy yellow to yellowish green, and gradually concentrated around the center over time (Figure 2). The fungus isolated from the koji starter 'MC-01' was identified as *A. oryzae* strain BJ-1 (NCBI GenBank accession no. MT305999, E-Value: 0.0, identity: 100%). *A. oryzae*, commonly known as koji mold, is the predominant koji-making fungus used in soy sauce production. *A. oryzae* is a filamentous fungus that plays a crucial role in the fermentation process of soy sauce production [29] and is responsible for the breakdown of proteins, carbohydrates, and lipids in raw materials, resulting in the distinctive flavor and aroma of soy sauce.

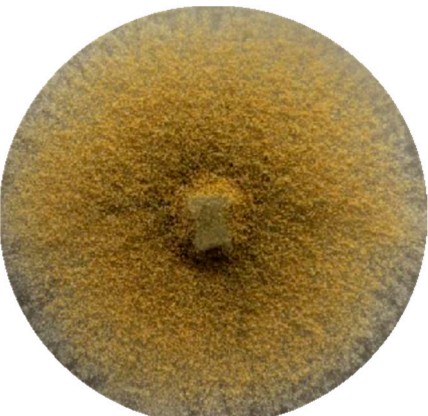

**Figure 2.** Morphological characterization of *Aspergillus oryzae* MC-01 cultured on a Potato Dextrose Agar plate. The strain was cultured at 28 °C for three days.

### 3.2. Proximate Compositions of Defatted Soybean Meal and Wheat

The proximate compositions of defatted soybean meal and wheat are listed in Table 1. Defatted soybean meal contained approximately 9.97% moisture, whereas wheat contained 10.64% moisture. Defatted soybean meal had a higher crude protein content (42.99%) than that of wheat (10.56%). Soybeans are well known for their high protein content and are considered an excellent source of plant-based proteins. Defatted soybean meal contained 2.73% crude fat, whereas wheat had a lower crude fat content (1.69%). As implied by the term "defatted", fat had been removed from the soybean meal. Defatting involves the application of pressure, which can disrupt the cell membrane and significantly affect the tissues of soybean, resulting in more efficient enzymatic action in defatted soybean meal owing to the enhanced penetration of enzymes. Defatted soybean meal contained 5.67% crude ash, whereas wheat contained 1.55%. The mineral content of soybeans, including defatted soybean meal, was approximately 5%, which is consistent with the findings of a previous study [6]. Carbohydrate content was approximately 39.74% in defatted soybean meal compared to 75.46% in wheat. Consequently, defatted soybean meal is a rich source of nitrogen, whereas wheat has a high carbon content, which is crucial for microbial growth. Wheat has traditionally been used as a carbon source for microbial growth during soy sauce brewing, and the mixing of roasted wheat with a low moisture content and steamed defatted soybean meal with a high moisture content has been reported to prevent bacterial growth [4].

**Table 1.** Proximate compositions of defatted soybean meal and wheat (%).

| Parameters | Defatted Soybean | Wheat |
|---|---|---|
| Moisture | 9.97 ± 3.34 | 10.64 ± 0.42 |
| Crude protein | 42.99 ± 0.12 | 10.56 ± 0.70 |
| Crude fat | 2.73 ± 0.01 | 1.69 ± 0.041 |
| Crude ash | 5.67 ± 0.21 | 1.55 ± 0.11 |
| Carbohydrate * | 39.74 ± 4.37 | 75.46 ± 0.87 |

Values are presented as the mean ± SD (*n* = 3); * Carbohydrate = 100 − (moisture + crude protein + crude fat + crude ash).

### 3.3. Optimization of Medium Composition for Liquid Starters

The mycelial content in the medium increased with an increase in substrate (soybean meal and wheat) concentrations (Table 2). The mycelium concentration increased in all samples, reaching a maximum in T3 at 17.95 mg/mL at 96 h. T3 consistently exhibited the highest mycelium concentration at all time points, indicating faster mycelium growth compared to the other samples. Notably, the mycelial content increased rapidly from 0 to 24 h, except in the control, suggesting that the mycelia preferentially utilized defatted

soybean meal and wheat as nutritional sources for growth-free amino acids and glucose from either soybean meal or wheat in liquid staters. Carbon and protein are crucial nutrients for microbial growth, and *A. oryzae* may produce enzymes for the hydrolysis of proteins and carbohydrates from soybean and wheat to support growth under nutrient-limited conditions. The control group had the highest glucose concentration of 20.19 g/L, which gradually decreased (Table 2). The initially high glucose concentration in the control group could be attributed to the composition of PDB medium. In contrast, T1, T2, and T3 showed significant reductions in glucose concentrations from their initial levels, with T1 showing the lowest glucose concentration among all samples at most time points. This decrease in glucose concentration was likely due to mycelial growth. The control group had the lowest initial pH (4.35), which gradually increased thereafter (Table 2). T2 and T3 showed an increase in pH compared with the control, whereas T1 exhibited a higher pH compared with all other samples at most time points. This increase in pH can be attributed to the release of alkaline byproducts during fungal growth. The control initially had the highest acidity at 0.07%, which gradually decreased, following a trend similar to that of the pH changes (Table 2). T1 displayed a moderate increase in acidity compared to that of the control, whereas T2 and T3 showed a significant increase in acidity, indicating higher acid production. The control and all samples (T1, T2, and T3) initially showed no detectable enzyme activities for protease, $\alpha$-amylase, and glutaminase (Table 2); however, all three samples (T1, T2, and T3) displayed significant enzymatic activity at 24 h. This significant increase in enzyme activity suggests that mycelia actively produce these enzymes, potentially contributing to the breakdown of fungal substrates. Overall, T2 and T3 exhibited higher mycelial growth, higher enzyme activity, and more significant changes in pH and acidity than those of the control and T1 treatment. These results indicate that T2 and T3 are more effective in promoting the growth and enzymatic activity of mycelia, making them potentially more favorable candidates for koji production, considering that the enzymatic activity of *A. oryzae* is crucial for the quality and taste of the final soy sauce product. In this study, T2 (a 10% defatted soybean meal and wheat mixture at a ratio of 55:45) was used as a liquid starter for koji production.

**Table 2.** Microbial growth on medium compositions.

| Parameters | Samples * | Incubation Time (h) | | | | |
|---|---|---|---|---|---|---|
| | | 0 | 24 | 48 | 72 | 96 |
| Mycelium (mg/mL) | Control | 0.50 ± 0.00 [a] | 2.26 ± 0.00 [d] | 4.77 ± 0.02 [d] | 5.04 ± 0.06 [d] | 5.31 ± 0.16 [d] |
| | T1 | 0.50 ± 0.00 [a] | 7.03 ± 0.11 [c] | 11.47 ± 0.02 [c] | 13.18 ± 0.03 [c] | 14.05 ± 0.05 [b] |
| | T2 | 0.50 ± 0.00 [a] | 9.38 ± 0.03 [b] | 12.37 ± 0.02 [b] | 15.04 ± 0.05 [b] | 16.47 ± 0.03 [c] |
| | T3 | 0.50 ± 0.01 [a] | 13.04 ± 0.05 [a] | 13.94 ± 0.06 [a] | 16.55 ± 0.05 [a] | 17.95 ± 0.05 [d] |
| Glucose (g/L) | Control | 20.19 ± 0.75 [a] | 15.93 ± 0.10 [a] | 10.13 ± 0.11 [a] | 11.64 ± 0.06 [a] | 9.75 ± 0.22 [a] |
| | T1 | 1.49 ± 0.01 [b] | 0.02 ± 0.01 [d] | 0.04 ± 0.01 [c] | N.D. ** | N.D. |
| | T2 | 1.62 ± 0.01 [b] | 0.19 ± 0.01 [c] | 0.29 ± 0.01 [b] | 0.20 ± 0.01 [c] | 0.09 ± 0.01 [bc] |
| | T3 | 1.84 ± 0.01 [b] | 0.32 ± 0.01 [b] | 0.21 ± 0.01 [b] | 0.36 ± 0.01 [b] | 0.23 ± 0.01 [b] |
| pH | Control | 4.35 ± 0.00 [c] | 4.45 ± 0.01 [d] | 4.54 ± 0.01 [d] | 4.88 ± 0.01 [d] | 5.09 ± 0.01 [d] |
| | T1 | 6.26 ± 0.01 [a] | 5.46 ± 0.01 [c] | 5.97 ± 0.01 [a] | 7.01 ± 0.01 [a] | 7.59 ± 0.01 [a] |
| | T2 | 6.26 ± 0.01 [a] | 5.66 ± 0.01 [b] | 5.49 ± 0.01 [c] | 5.93 ± 0.01 [b] | 5.70 ± 0.01 [b] |
| | T3 | 6.25 ± 0.01 [b] | 5.80 ± 0.01 [a] | 5.77 ± 0.01 [b] | 5.32 ± 0.01 [c] | 5.50 ± 0.01 [c] |
| Acidity (%) | Control | 0.07 ± 0.00 [a] | 0.06 ± 0.00 [d] | 0.05 ± 0.00 [d] | 0.05 ± 0.00 [d] | 0.05 ± 0.00 [d] |
| | T1 | 0.06 ± 0.00 [c] | 0.07 ± 0.00 [c] | 0.09 ± 0.00 [c] | 0.10 ± 0.00 [c] | 0.11 ± 0.00 [c] |
| | T2 | 0.06 ± 0.00 [bc] | 0.08 ± 0.00 [b] | 0.28 ± 0.00 [b] | 0.25 ± 0.00 [b] | 0.24 ± 0.00 [b] |
| | T3 | 0.06 ± 0.00 [b] | 0.09 ± 0.00 [a] | 0.35 ± 0.00 [a] | 0.51 ± 0.01 [a] | 0.50 ± 0.00 [a] |

**Table 2.** *Cont.*

| Parameters | Samples * | Incubation Time (h) | | | | |
|---|---|---|---|---|---|---|
| | | 0 | 24 | 48 | 72 | 96 |
| Protease activity (mUnit/mL) | Control | 0.00 ± 0.00 | 0.00 ± 0.00 [d] | 0.00 ± 0.00 [d] | 0.00 ± 0.00 [d] | 0.00 ± 0.00 [d] |
| | T1 | 0.00 ± 0.00 | 16.67 ± 0.10 [b] | 17.22 ± 0.04 [b] | 17.45 ± 0.06 [a] | 13.58 ± 0.04 [c] |
| | T2 | 0.00 ± 0.00 | 21.25 ± 0.05 [a] | 23.21 ± 0.12 [a] | 10.57 ± 0.08 [c] | 19.12 ± 0.44 [a] |
| | T3 | 0.00 ± 0.00 | 10.34 ± 0.07 [c] | 6.15 ± 0.14 [c] | 11.70 ± 0.07 [b] | 16.13 ± 0.06 [b] |
| α-Amylase activity (unit/mL) | Control | 0.00 ± 0.00 | 0.00 ± 0.00 [d] | 0.00 ± 0.00 [c] | 135.69 ± 9.50 [c] | 160.27 ± 16.98 [c] |
| | T1 | 0.00 ± 0.00 | 95.30 ± 0.71 [c] | 187.83 ± 5.29 [b] | 171.51 ± 3.54 [b] | 168.38 ± 2.22 [b] |
| | T2 | 0.00 ± 0.00 | 167.99 ± 2.64 [b] | 224.50 ± 3.44 [a] | 214.29 ± 4.97 [a] | 185.10 ± 4.01 [a] |
| | T3 | 0.00 ± 0.00 | 199.05 ± 3.52 [a] | 100.48 ± 2.37 [c] | 99.62 ± 4.35 [d] | 90.63 ± 1.88 [d] |
| Glutaminase (unit/mL) | Control | 0.00 ± 0.00 | 8.53 ± 0.04 [d] | 12.31 ± 0.03 [d] | 13.60 ± 0.51 [d] | 22.87 ± 0.20 [d] |
| | T1 | 0.00 ± 0.00 | 12.92 ± 0.04 [c] | 22.95 ± 0.06 [c] | 19.45 ± 0.10 [c] | 24.17 ± 0.09 [c] |
| | T2 | 0.00 ± 0.00 | 24.18 ± 0.08 [b] | 30.61 ± 1.45 [b] | 41.95 ± 0.88 [b] | 59.83 ± 0.85 [a] |
| | T3 | 0.00 ± 0.00 | 55.37 ± 0.54 [a] | 39.51 ± 0.94 [a] | 74.80 ± 0.74 [a] | 52.43 ± 1.24 [b] |

The sample size was 1 g. Different superscripts (a to d) within the columns indicate significant differences ($p < 0.05$); * control was cultivated in Potato Dextrose Broth. The medium concentrations of defatted soybean meal and wheat (mixed at a ratio of 55:45) in T1, T2, and T3 were 5%, 10%, and 15%, respectively; ** not determined.

### 3.4. Characterization of Koji by Assessing Inoculation Ratio of Liquid Starter

The characteristics of koji according to the liquid starter inoculation rates are presented in Table 3. The moisture content of all samples was approximately 48–49% during the initial stage of fermentation, and a decrease in moisture content was observed in all samples after 72 h. The enzyme activities of each koji were limited after 48 h of incubation due to limited mycelial growth at lower moisture concentrations [6]. K20 exhibited the highest moisture content (46.87%). High moisture content may lead to the growth of undesirable bacteria present in the spores, which can be avoided by using a pure liquid culture medium. The initial pH values for all samples were within the range of 6.23–6.37, which changed during fermentation. All samples except the control showed a sharp decrease in pH at 48 h. Notably, pH rapidly increased in K1, K5, and K10, whereas it increased at 72 h in K20. Changes in pH are likely a result of metabolic activities of microorganisms that produce acidic or basic byproducts during fermentation. The pH of koji is an important process control parameter that exerts a considerable influence on the enzymatic reactions and proliferation of lactic acid bacteria and yeast during the Moromi stage. In particular, maintaining a pH of at least 6.6 in koji is essential to support the growth of lactic acid bacteria and ensure optimal nitrogen utilization and glutamic acid production during the Moromi stage [6]. In this study, all koji, except K20, were suitable for soy sauce production because of the attainment of pH > 6.6. The initial glucose content increased with an increase in the inoculum size. The glucose concentration increased during fermentation. K5 exhibited the highest glucose content at 72 h (24.43 g/L). These fluctuations in glucose content are likely a result of the utilization of glucose as a substrate by microorganisms during fermentation. Therefore, this phenomenon is considered to be associated with the secretion of α-amylase by microorganisms using glucose as a carbon source. The control, K1, and K5 groups initially showed low α-amylase activity, which increased significantly after 24 h. In particular, K5 showed the highest α-amylase activity at 72 h, which may be attributed to glucose concentration. K10 and K20 exhibited the highest α-amylase activity at 0 h, and this activity continued to increase during fermentation. As α-amylase breaks down starch into sugars, the increased activity suggests enhanced starch degradation in the samples. Glutaminase activity was not detected at 0 h in any of the samples. However, it was detected at 24 h in all samples, with the control and K20 samples showing high activity. This indicates that it catalyzed the production of glutamic acid from the amino acid glutamine during fermentation. Glutaminase acts exclusively in the presence of free glutamine. Similar to glutaminase, protease activity was not detected at 0 h in any sample;

however, it was detected in all samples at 24 h. Notably, K10 exhibited the highest activity at all pH values tested (pH 5.0, 7.0, and 8.0). Protease activity indicates proteolysis during fermentation, and takes precedence over glutaminase activity during soy sauce production. This is because the enhancement of glutamic acid content relies on protein hydrolysis, which is a prerequisite for glutaminase activity. The activities of all enzymes were more pronounced in koji utilizing liquid starters than those in the control. These findings suggest that K10 (10% inoculum of liquid starter) is particularly suitable for soy sauce production owing to the favorable characteristics of koji.

**Table 3.** Characteristics of koji according to the inoculation rate of liquid starter.

| Parameters | Samples * | Incubation Time (h) | | | |
|---|---|---|---|---|---|
| | | 0 | 24 | 48 | 72 |
| Moisture content ** (%) | Control | 48.18 ± 0.03 [d] | - | - | 40.23 ± 0.12 [b] |
| | K1 | 48.18 ± 0.01 [d] | - | - | 36.07 ± 0.06 [e] |
| | K5 | 48.52 ± 0.02 [c] | - | - | 38.43 ± 0.06 [d] |
| | K10 | 49.23 ± 0.01 [b] | - | - | 38.67 ± 0.06 [c] |
| | K20 | 49.43 ± 0.16 [a] | - | - | 46.87 ± 0.12 [a] |
| pH | Control | 6.37 ± 0.01 [a] | 6.25 ± 0.00 [b] | 6.68 ± 0.01 [a] | 7.07 ± 0.03 [a] |
| | K1 | 6.37 ± 0.00 [a] | 6.29 ± 0.01 [a] | 5.89 ± 0.00 [b] | 6.73 ± 0.01 [c] |
| | K5 | 6.23 ± 0.01 [d] | 6.12 ± 0.00 [c] | 5.24 ± 0.00 [d] | 6.95 ± 0.01 [b] |
| | K10 | 6.34 ± 0.01 [b] | 6.10 ± 0.00 [d] | 5.17 ± 0.00 [e] | 6.97 ± 0.01 [b] |
| | K20 | 6.28 ± 0.01 [c] | 5.98 ± 0.00 [e] | 5.46 ± 0.00 [c] | 5.66 ± 0.02 [d] |
| Glucose content (g/L) | Control | 0.36 ± 0.01 [e] | 3.61 ± 0.02 [d] | 10.63 ± 0.55 [c] | 21.23 ± 0.15 [b] |
| | K1 | 0.49 ± 0.00 [d] | 6.21 ± 0.02 [c] | 9.11 ± 0.04 [d] | 19.23 ± 0.06 [c] |
| | K5 | 1.02 ± 0.04 [c] | 6.62 ± 0.05 [b] | 15.30 ± 0.30 [b] | 24.43 ± 0.50 [a] |
| | K10 | 1.71 ± 0.01 [b] | 6.56 ± 0.00 [b] | 17.40 ± 0.00 [a] | 21.73 ± 0.25 [b] |
| | K20 | 3.41 ± 0.03 [a] | 10.33 ± 0.15 [a] | 15.60 ± 0.44 [b] | 13.47 ± 0.49 [d] |
| α-Amylase activity (unit/mL) | Control | 22.33 ± 0.58 [d] | 63.67 ± 0.58 [e] | 6570.00 ± 115.33 [d] | 9210.00 ± 5.57 [d] |
| | K1 | 1.83 ± 0.76 [e] | 497.67 ± 4.04 [d] | 3769.67 ± 60.58 [e] | 10,416.33 ± 29.16 [bc] |
| | K5 | 25.33 ± 0.76 [c] | 985.67 ± 4.04 [c] | 6936.33 ± 84.56 [c] | 10,603.67 ± 300.07 [ab] |
| | K10 | 60.43 ± 0.51 [b] | 1044.33 ± 33.56 [b] | 8033.00 ± 22.52 [b] | 10,813.00 ± 32.51 [a] |
| | K20 | 69.90 ± 1.15 [a] | 4832.00 ± 22.07 [a] | 9181.00 ± 16.82 [a] | 10,179.67 ± 38.21 [c] |
| Glutaminase (unit/mL) | Control | N.D. *** | 322.33 ± 1.15 [b] | 520.33 ± 10.50 [b] | 1296.33 ± 31.64 [a] |
| | K1 | N.D. | 156.33 ± 1.53 [e] | 363.67 ± 12.22 [d] | 675.00 ± 8.66 [d] |
| | K5 | N.D. | 218.33 ± 1.15 [d] | 373.67 ± 2.52 [cd] | 725.00 ± 3.00 [d] |
| | K10 | N.D. | 284.00 ± 1.73 [c] | 385.67 ± 5.86 [c] | 965.33 ± 5.69 [c] |
| | K20 | N.D. | 402.00 ± 8.19 [a] | 812.67 ± 1.15 [a] | 1099.00 ± 59.81 [b] |
| Acidic protease activity (pH 5.0) | Control | N.D. | 155.33 ± 2.31 [c] | 2456.33 ± 58.48 [b] | 2667.00 ± 59.91 [a] |
| | K1 | N.D. | 101.00 ± 2.65 [d] | 1748.67 ± 61.85 [c] | 2636.67 ± 77.69 [a] |
| | K5 | N.D. | 188.00 ± 7.00 [c] | 2464.00 ± 29.55 [b] | 2663.33 ± 54.99 [a] |
| | K10 | N.D. | 454.00 ± 10.00 [b] | 2421.00 ± 49.43 [b] | 2685.33 ± 64.27 [a] |
| | K20 | N.D. | 2033.33 ± 59.50 [a] | 2762.00 ± 75.74 [a] | 2652.00 ± 112.21 [a] |
| Neutral protease activity (pH 7.0) | Control | N.D. | 110.33 ± 3.06 [c] | 2634.33 ± 39.88 [c] | 2595.33 ± 4.04 [d] |
| | K1 | N.D. | 57.00 ± 1.00 [c] | 1949.67 ± 5.51 [e] | 2522.00 ± 0.00 [e] |
| | K5 | N.D. | 136.07 ± 115.95 [c] | 2571.33 ± 0.58 [d] | 2680.33 ± 0.58 [c] |
| | K10 | N.D. | 442.67 ± 2.52 [b] | 2682.67 ± 11.02 [b] | 2776.33 ± 1.53 [a] |
| | K20 | N.D. | 2146.67 ± 38.81 [a] | 2752.00 ± 3.00 [a] | 2711.33 ± 11.50 [b] |
| Alkaline protease activity (pH 8.0) | Control | N.D. | 165.00 ± 4.58 [d] | 2699.33 ± 11.50 [c] | 2821.33 ± 8.96 [ab] |
| | K1 | N.D. | 110.33 ± 0.58 [e] | 1932.00 ± 13.53 [e] | 2769.67 ± 60.58 [b] |
| | K5 | N.D. | 228.00 ± 1.00 [c] | 2692.00 ± 13.00 [d] | 2860.00 ± 4.58 [a] |
| | K10 | N.D. | 479.33 ± 4.51 [b] | 2741.00 ± 7.00 [b] | 2814.33 ± 6.66 [ab] |
| | K20 | N.D. | 2258.67 ± 8.14 [a] | 2971.67 ± 25.58 [a] | 2817.67 ± 7.77 [ab] |

Sample size was 1 g. Different superscripts (a to e) within the columns indicate significant differences ($p < 0.05$); * control; ** moisture content measured at the start and end of koji-making process; *** not determined.

## 4. Conclusions

Microorganisms confer a desirable taste, aroma, and color to soy sauce, which is used as a basic condiment and an all-purpose seasoning in cuisines of several Asian countries [30]. In fact, koji with higher enzyme activity, such as protease activity, is closely associated with higher final soy sauce quality (soluble amino acid and organic acid content). In this study, we investigated the feasibility of using liquid culture inoculum for koji production during soy sauce manufacturing. Notably, the quality of koji prepared using a liquid stater was comparable to that of koji prepared using spores. *A. oryzae* BJ-1, which is crucial for soy sauce fermentation, was isolated from a commercial koji starter. The proximate compositions of defatted soybean meal and wheat, which are key ingredients in soy sauce production, were examined to better understand their nutritional profiles. This study focused on optimizing the medium composition of liquid starters. Mycelial content increased with higher inducer concentrations. Mycelial growth was rapid, suggesting the efficient utilization of defatted soybean meal and wheat as nutritional sources. Enzymatic activities, including those of protease, α-amylase, and glutaminase, were considerably enhanced in all liquid starters. These findings suggest that liquid starters improve mycelial growth and enzyme production, which are critical for high-quality soy sauce production. Additionally, koji produced by varying the inoculation rate of the liquid starter was characterized. Koji samples displayed varying moisture content, pH values, glucose concentrations, and enzymatic activities depending on the inoculation rate. The liquid starter inoculum (10%) showed favorable characteristics for soy sauce production, particularly in terms of α-amylase and protease activity, which are essential for starch and protein degradation.

These results suggest that the use of liquid culture inoculum for koji production holds promise as an efficient and controllable approach to low-salt soy sauce manufacturing, with the elimination of safety hazards associated with spores during koji fermentation. This approach could potentially reduce bacterial contamination, improve mycelial growth, enhance enzymatic activity, and ultimately contribute to the production of high-quality soy sauce. This study lays the foundation for further exploration and development of liquid culture-based methods in the soy sauce industry, potentially leading to safer and more efficient production processes and the production of salt-free fermented soy sauce, which could have major applications in the food seasoning industry. Future studies should investigate and refine the use of liquid culture inocula in large-scale soy sauce production. Microbial activities play an important role in the fermentation process, as they alter the moisture content, pH, glucose consumption, and activities of enzymes such as α-amylase, glutaminase, and proteases. These findings provide valuable insights into the metabolic dynamics of fermentation and contribute to our understanding of substrate bioconversion by microorganisms under different conditions.

**Author Contributions:** Conceptualization, methodology, investigation, formal analysis, data curation, writing—original draft, J.C.; conceptualization, methodology, investigation, validation, data curation, writing—original draft, visualization, writing—review and editing, T.K.; methodology, investigation, formal analysis, Y.P.; conceptualization, validation, writing—review and editing, supervision, project administration, funding acquisition, A.Y.K. All authors have read and agreed to the published version of the manuscript.

**Funding:** This work was supported by the faculty research fund of Sejong University in 2022.

**Institutional Review Board Statement:** Not applicable.

**Informed Consent Statement:** Not applicable.

**Data Availability Statement:** The data presented in this study are available on request from the corresponding author.

**Conflicts of Interest:** The authors declare that they have no competing financial interests or personal relationships that may have influenced the work reported in this study.

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
