# Peer review of "Characteristics of Koji Using Liquid Starter for Soy Sauce Production"

_fermentation, doi:10.3390/fermentation9110979_

Round 1
Reviewer 1 Report
Comments and Suggestions for Authors
The hypothesis in the final section of the Introduction is not convincing. Trying to avoid contamination by having a pure culture in a closed environment is not equal to moving from solid to liquid state fermentation, which is the focus of this study. If the addition of salt is to kill the fungal culture (line 36), so that the ripening can continue to occur, then how is salt-free fermentation which allows continued fungal growth going to be successful?
The format studied for submerged koji fermentation has major differences from the traditional solid-state mode, making it not possible to compare teh two. These include the defatting of the soybean, grinding of the solid substrates, and high inoculum ratio in mycelial form in this study.
Figure 1 doesn't seem to be what the text (line 185) is referring to.
Missing reference in line 228.
There is some discussion about the suitability of the liquid fermentation for soy sauce production but mostly postulative since it was not actually implemented in soy sauce fermentation. What is the significance of the result values, especially in section 3.4? Comparison to traditional or solid-state fermentation data will make the case more convincing.
Comments on the Quality of English LanguageFrom section 3.4 onwards, it is difficult to comprehend what the authors are trying to express, due to poor language and lacking information.
The remainder of the manuscript contains several minor grammatical errors which need to be corrected.
Author Response
Subject: Revision of the manuscript
Dear,
Thank you for giving me the opportunity to submit a revised draft of my manuscript titled “Characteristics of Koji Using Submerged Culture for Soy Sauce Production” (fermentation-2608931) to Fermentation (ISSN 2311-5637). We appreciate the time and effort that you and the reviewers have dedicated to providing your valuable feedback on our manuscript. We are grateful again to the reviewers for their insightful comments on our paper. It is our belief that the manuscript is substantially improved after making the suggested edits.
Our responses that following the editor and reviewer comments are given in a point-by-point manner below. We have marked the changes in red within the manuscript. The revision has been developed in consultation with all coauthors, and each author has given approval to the final form of this revision.
We hope the revised version is now suitable for publication and look forward to hearing from you in due course.
Sincerely yours,

Reviewer 2 Report
Comments and Suggestions for Authors
The quality of the manuscript is good, and it will be beneficial to the scientific community which could be used as a guide into the area of research to be explored.
The manuscript is well written, and it will be understood by the general public.
There are a few recommendations that could be added to enhance quality, clarity and readability.
The methodology: The major novelty of this article is the new design of liquid starter. I suggest building a flow chart, may be with some photos were taking during the experiment, to explain the two different starter cultures and koji fermentation processes. It may also show the sampling time and different tests.
Fermentation time and Sensory quality of the final products are key points in most of the food/beverage fermentation processes. I suggest the authors to highlight these points in the discussion (how the koji using liquid stater might accelerate the subsequent fermentation process (brine maturation) and impact the final product quality. If the authors have not enough data/thoughts about them in these current experiments, kindly refer to them for future studies under the conclusion.
General points
Defatted soybean, do you mean soybean meal? If yes kindly mention it
Line 36? Kindly add this statement ‘…in addition, semi anaerobic environment created underwater, as well as high salt content cause the decline of fungal growth during Mormori’. Please find more about that in the below listed references.
Line 40. I think a brief discussion about other microbial groups which lead the fermentation during Mormori maturation stage (LAB, other acid tolerant bacteria, yeasts), and how the degrade substates created by the fungal enzymes created during Koji will be used to support the growth of these microbial groups in the subsequent fermentation.
‘Microbial succession and the functional potential during the fermentation of Chinese soy sauce brine. Front. Microbiol. 5:556. doi: 10.3389/fmicb.2014.00556. By Sulaiman J, et al. (2014)
‘Soybean fermentation: Microbial ecology and starter culture technology, Critical Reviews in Food Science and Nutrition, DOI: 10.1080/10408398.2023.2188951’ by Elhalis, et al. (2023)
Line 102: what is the total weight?
Line 112: again, please mention the sample size (how many replicates)
Line 126: with spores? Please rewrite the spore origin (Koji starter ‘MC-01). Also illustrate the sample number (how many replicates per each fermentation type). Please review all methods with sample size.
Line 145 ‘Total endo-protease activity’’ why endo protease?
Line 214, Please correct ..defatted
Line 288: Please add a reference.
Table 2 and 3. Please write the sample size.
Comments on the Quality of English LanguageThe manuscript is well written.
Author Response

(The authors gave the same response as above.)

Reviewer 3 Report
Comments and Suggestions for Authors
Dear authors,
It is very interesting that you propose liquid starters to change spores inoculation for the production of the traditional soy sauce. There are some aspects and recommendations to be considered which are listed below:
I am not sure if the title could be changed to “Characteristics of Koji Using liquid starters for Soy Sauce Production”
Line 19. In which characteristics?. Specify.
Line 43. Include temperatura and hymidity range
Line 43. Indicate where koji is normally produced
Line 66-78. Please, rewrite this paragraph in a way that the reader can undertand exactly what is performed in the present study.
Line 87. Could you explain why the liquid media was ground?
Line 90. A full stop after “study” is missing.
Line 101-102. In this phrase I understood that you mixed soybean with wheat in a proportion of 55:45, by using a solid/liquid proportion of 5 %, 10 % and 15%. If I am right, in which solution was suspended the solid?
Line 107. Which volumen of media was taken for dry weight analysis?
Line 108. Why were the solids dried?
Line 120. Amount
Line 138. What the pH of the starch solution adjusted?
Line 147. Substrate in lower case
Line 147. Which type of ethanol?
Line 153. Please, could you explain the origin of protease activity determination?. Is it published in another paper?
Line 159. When you mentione “the enzyme was added”, do you refer to your simple?
Line 167 and 172. Please, reference the enzymes methods as you did for acidic protease activity
Line 178. Ammonia in lower case
Line 184. What do you mean with “. 0.2 mL of Nessler's 183 reagent was added to the color?”
Line 185. You describe that an equation was used to determine glutaminase activity ,shown in figure 1. However, in figure 1 a photo of A. oryzae is shown.
Line 214. Change to “defated soybean..”
Line 216. Why were used defated raw materials?
Line 226. Do you mean 39.74 % of carbohydrates?
Line 228. How could you explain the prevention of bacterial growth?
Line 251. Do you know which alkaline by-products are produced?
Line 265. Please, could you better explain the selection of T2?
Line 274. This phrase “There cannot be obtained koji with higher enzyme activities because the growth of bacteria is insufficient when the lower moisture condition” is not clear
Line 298. Include a full stop after “during fermentation”
Table 3. Why pH increased at time 72 h?
Line 311-312. You mentioned “ These findings suggested that K10 (10% inoculum of liquid starter) is particularly suitable for soy sauce production due to its favorable koji characterization”. You should include a deep explanation to select this condition. Which characteristics are expected to be appropiate to achieve a better say sauce.
Line 317. Change to conclusions. Conclusion section is very poor
Author Response

(The authors gave the same response as above.)

Round 2
Reviewer 1 Report
Comments and Suggestions for Authors
On the whole, it seems that the reported results indicate that the liquid starter culture method is not worse off than the traditional spore inoculation. The liquid culture here may be of higher purity, but spores of the same purity can also be prepared. Other criteria including koji pH, glucose, and enzyme activity are similar to the control. Hence, it is not clear what is the advantage of the proposed strategy over the traditional approach.

The manuscript still has several grammatical minor issues, as well as a couple of sentences with unclear meaning.
The methodology regarding the experimental design could be made clearer e.g. explaining how each control was done.
Author Response
Thank you for reveiwer comments.
The advantage of the proposed strategy over the traditional approach are
1) Potential makes lower salt soy sauce
2) Safety of soy sauce due to prevent contain mycotoxin
3) Wokers safety from spore handing.
We have marked the changes in red within the manuscript. The revision has been developed in consultation with all coauthors, and each author has given approval to the final form of this revision.